# Chain-of-Thought Embeddings for Stance Detection on Social Media

**Joseph Gatto, Omar Sharif, Sarah M. Preum**
Department of Computer Science, Dartmouth College
{joseph.m.gatto.gr, omar.sharif.gr, sarah.masud.preum} @ dartmouth.edu

## Abstract

Stance detection on social media is challenging for Large Language Models (LLMs), as emerging slang and colloquial language in online conversations often contain deeply implicit stance labels. Chain-of-Thought (COT) prompting has recently been shown to improve performance on stance detection tasks — alleviating some of these issues. However, COT prompting still struggles with implicit stance identification. This challenge arises because many samples are initially challenging to comprehend before a model becomes familiar with the slang and evolving knowledge related to different topics, all of which need to be acquired through the training data. In this study, we address this problem by introducing **COT Embeddings** which improve COT performance on stance detection tasks by *embedding* COT reasonings and integrating them into a traditional RoBERTa-based stance detection pipeline. Our analysis demonstrates that 1) text encoders can leverage COT reasonings with minor errors or hallucinations that would otherwise distort the COT output label. 2) Text encoders can overlook misleading COT reasoning when a sample's prediction heavily depends on domain-specific patterns. Our model achieves SOTA performance on multiple stance detection datasets collected from social media.

## 1 Introduction

Detecting the stance of a text with respect to a certain topic is vital to many NLP tasks (Hardalov et al., 2022). Detecting stances on social media platforms like Twitter poses unique challenges, as emerging knowledge and colloquial language patterns can make it difficult to detect stances without additional context. For example, consider the top tweet shown in Figure 1. This tweet contains no direct mention of Donald Trump, and is thus difficult to classify without further context — such as how Trump supporters on Twitter widely supported voter fraud propaganda. Such emerging knowledge

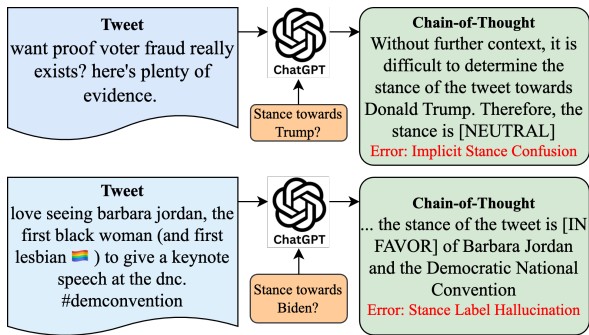

Figure 1: Common errors made by Chain-of-Thought reasoning models. *Implicit Stance Confusion* refers to LLMs inability to understand the implicit reference to the stance topic. In the example above, ChatGPT should have predicted that the tweet is [IN FAVOR] of Trump. In this context, *Stance Label Hallucination* refers to the scenario where LLMs use the label space to argue the wrong point. In this example, the reasoning is correct, but ChatGPT used the [IN FAVOR] label towards the wrong topic.

is difficult for LLMs with knowledge cut-offs to understand and may only be discernible by observing similarly labeled samples in the training set.

One way to solve this problem is by employing models with extensive world knowledge. For example, recent works have shown that using ChatGPT on Stance Detection can provide significant performance increases (Zhang et al., 2023a,b). Unfortunately, LLMs (e.g., ChatGPT, Llama) still have many issues understanding complex stance relationships from Twitter data. In this study, we highlight two issues with the state-of-the-art **Chain-of-Thought (COT)** prompting approach to stance detection. 1) *Implicit Stance Confusion:* As shown in Figure 1, LLMs continue to struggle with understanding implicit tweet stance, even when employing advanced prompting strategies like COT reasoning (Wei et al., 2023). 2) *Stance Label Hallucination:* LLMs are prone to hallucinations, which cause them to output sound reasonings, but for the wrong stance topic (see Figure 1 example). Even

when LLMs analyze the correct topic, they are also prone to using the provided label space incorrectly, producing accurate but ill-structured outputs.

In this study, we mitigate these two problems by introducing **Chain-of-Thought (COT) Embeddings**. Our approach feeds the COT reasoning text to a transformer encoder to be used as an additional feature in a traditional stance detection pipeline. The intuition behind this approach is three-fold: *(i)* Text encoders are robust to stance label hallucinations if the COT reasoning is correct. This can make incorrect COT predictions useful in a text classification pipeline. *(ii)* Text encoders can choose to ignore certain signals as needed. Thus, when a sample is too implicit to be understood by LLMs, the model may choose to focus on how similar tweets were classified. *(iii)* COT reasonings can inject world knowledge into a text encoder. That is, COT texts often contain reasonings and justifications grounded in world knowledge not available in the tweet. We find that, by using this approach, we can achieve state-of-the-art results on multiple stance detection datasets.

A summary of our contributions is as follows:

1. To the best of our knowledge, this is the first investigation into the embedding of COT reasonings. Our approach achieves state-of-the-art results on two stance detection datasets: Tweet-Stance (Mohammad et al., 2016; Barbieri et al., 2020) and Presidential-Stance (Kawintiranon and Singh, 2021).

2. Our error analysis on COT reasoning highlights two key flaws on stance detection tasks: *Implicit Stance Confusion* and *Stance Label Hallucinations*. Our approach, Chain-of-Thought Embeddings, makes COT outputs more robust to these two issues.

## 2  Related Work

**Stance Detection:**  This task is a well-explored research problem, where early studies employed various machine learning and deep learning techniques (Hardalov et al., 2022). The emergence of large language models has further pushed the state-of-the-art performance on many stance detection datasets (Li and Caragea, 2021). Many stance detection problems require domain-specific solutions with models which explicitly inject world knowledge into stance detection systems (He et al., 2022;

| | Tweet-Stance | | | | | Presidential-Stance | |
| --- | --- | --- | --- | --- | --- | --- | --- |
| | HC | FM | LA | AT | CC | BD | TR |
| Train | 620 | 597 | 587 | 461 | 355 | 875 | 875 |
| Dev | 69 | 67 | 66 | 52 | 40 | - | - |
| Test | 295 | 285 | 280 | 220 | 169 | 375 | 375 |
| *Class-wise distribution of topics* | | | | | | | |
| Neutral | 256 | 170 | 167 | 145 | 203 | 487 | 410 |
| Against | 565 | 511 | 544 | 464 | 26 | 385 | 499 |
| Favor | 163 | 268 | 222 | 124 | 335 | 378 | 341 |
| Total | 984 | 949 | 933 | 733 | 564 | 1250 | 1250 |

Table 1:  Topic-wise (e.g., HC, FM, TR) distribution of train, development, test, and classes of the Tweet-Stance and Presidential-Stance datasets. The Presidential-Stance dataset does not have a development set.

Liu et al., 2021). This work is motivated by knowledge infusion but substantially differs from existing works. To the best of our knowledge, while some prior work has used prompting for stance detection (Zhang et al., 2023b), no work has attempted to use LLMs as a knowledge base for improved stance detection. While we also do not explicitly explore LLMs as a knowledge extraction tool, we do find that our method has the capacity to inject world knowledge into a inference pipeline due to the nature of COT text generation.

**LLMs for Stance Detection**  Recently, few works have used ChatGPT for stance detection directly Zhang et al. (2023a,b). In (Zhang et al., 2023b), the authors achieve superior performance on several stance detection datasets by prompting ChatGPT to do Chain-of-Thought inference. In this study, we use a similar prompting strategy to perform stance detection, but show the benefits of embedding these COT reasoning texts and using them as a feature in a stance detection pipeline.

## 3  Methods

We employ a 1-shot COT prompt for each tweet in each dataset, aiming to determine the stance of the tweet in relation to a specific topic[1]. We specifically ask the models to provide a COT reasoning and to include its predicted label in brackets (e.g. [NEUTRAL] for a neutral tweet), so the output may be parsed and converted to a numeric representation. An example tweet and corresponding

---

[1]Please refer to Appendix B for details on the prompts used for each task

|        | Favor | Against | Neutral |
|--------|-------|---------|---------|
| *Tweet-Stance* | | | |
| Train | 678 | 1254 | 688 |
| Dev | 75 | 141 | 78 |
| Test | 304 | 715 | 230 |
| *Presidential-Stance-Biden* | | | |
| Train | 266 | 279 | 330 |
| Test | 112 | 106 | 157 |
| *Presidential-Stance-Trump* | | | |
| Train | 243 | 347 | 285 |
| Test | 98 | 152 | 125 |

Table 2: Class-wise (i.e., neutral, against, favor) train, development, and test set statistics of the Tweet-Stance and Presidential-Stance datasets. Note that we aggregate the topics in Tweet-Stance in our experiments.

COT excerpt can be found in Figure 1.

After producing COT reasoning for a given text, we embed it with a transformer encoder and use it as a part of a stance detection pipeline. We specifically use a RoBERTa model (Liu et al., 2019) trained on Twitter data as our encoder since it has been shown to perform better on Tweet-Stance when compared to RoBERTa-base[2]. We denote this model as Twitter-RoBERTa (TR) in this paper.

We consider three different Twitter-RoBERTa variants in our experiments. **TR-Tweet**: We fine-tune with only tweet information. **TR-COT**: Fine-tune using only COT reasoning as the input and **TR-Tweet+COT**: Fine-tune Twitter-RoBERTa where tweet and COT reasoning are treated as a pair-wise input to the model (i.e. Tweet and COT reasoning texts are concatenated and jointly encoded by the pre-trained language model). All fine-tuning follows the standard text classification pipeline introduced in (Devlin et al., 2018). Please refer to Appendix A for model hyperparameters and training details for each stance detection task.

### 3.1 Dataset

We assess our method on two well-known Twitter-based stance detection datasets: Tweet-Stance (Mohammad et al., 2016; Barbieri et al., 2020) and Presidential-Stance (Kawintiranon and Singh, 2021). These datasets involve a 3-way classification task to determine whether tweets are in favor, against, or neutral towards a specific topic. The Tweet-Stance dataset comprises five topics: Hillary Clinton (HC), Feminism (FM), Abortion

(LA), Atheism (AT), and Climate Change (CC). The Presidential-Stance dataset contains two sub-tasks focusing on the 2020 election cycle, with annotation for stance towards presidential candidates Joe Biden (BD) and Donald Trump (TR). The topic-wise and class-wise distribution and statistics for the training, development, and test sets of both datasets are presented in Table 1 and Table 2, respectively. The class-wise distribution indicates that both datasets are skewed towards the *against* class.

### 3.2 Evaluation

**Tweet-Stance:** We report the macro average of the *Favor* and *Against* F1 scores as defined in (Barbieri et al., 2020). We report baseline performance of 3 encoder-based stance detection models: BERT-Spc (Devlin et al., 2018), BERT-GCN (Lin et al., 2021) and PT-HCL (Liang et al., 2022) as well as two ChatGPT prompting based methods: DQA and StSQA (Zhang et al., 2023b). All baseline scores are extracted from (Zhang et al., 2023b), where we note that evaluation was conducted on only a subset of the label space.

**Presidential-Stance:** We report both the per-class F1 score and the macro average F1 score as reported in (Kawintiranon and Singh, 2021). Due to the lack of development set in Presidential-Stance, we report the average results over three experimental trials with different random seeds. We report the results of three baseline models BERT (Devlin et al., 2018), SKEP (Tian et al., 2020), and KE-MLM (Kawintiranon and Singh, 2021).

## 4 Results

### 4.1 Tweet-Stance

Results on Tweet-Stance are exhibited in Table 3. Results show that TR-Tweet+COT produces the best-performing model on Tweet-Stance, with an F1 score of 76.3. Notably, we can retain most of the performance by only embedding the COT reasoning, as TR-COT has only a 0.6 difference in F1 from TR-Tweet+COT. Our best model provides a **6.1-pt improvement** over our ChatGPT COT reasoning model, and simply embedding COT provides a 5.5 boost in F1 vs extracting results from COT directly.

After investigating the subset of samples where TR-Tweet+COT is correct, but disagrees with the prediction from ChatGPT COT, we find that 74%

---

[2]Huggingface Model: cardiffnlp/twitter-roberta-base-sep2022

| Model | HC | FM | LA | AT | CC | $\text{F1}_{avg}$ |
|---|---|---|---|---|---|---|
| *Baselines* | | | | | | |
| BERT-Spc[†] | 49.6 | 41.9 | 44.8 | - | - | - |
| BERT-GCN[†] | 50.0 | 44.3 | 44.2 | - | - | - |
| PT-HCL[†] | 54.5 | 54.6 | 50.9 | - | - | - |
| DQA[†] | 78.0 | 69.0 | 59.3 | - | - | - |
| StSQA[†] | 78.9 | 68.7 | 61.5 | - | - | - |
| *ChatGPT Only* | | | | | | |
| 0-Shot | 71.5 | 61.6 | 49.1 | 21.6 | 37.1 | 51.6 |
| COT | 75.3 | 71.3 | 62.6 | 58.3 | 67.3 | 70.2 |
| *COT-Embeddings + Twitter-RoBERTa (TR)* | | | | | | |
| TR-Tweet | 59.0 | 56.6 | **64.0** | 67.0 | 52.6 | 69.0 |
| TR-COT | **81.3** | **72.6** | 61.4 | 70.7 | **69.3** | 75.7 |
| TR-Tweet +COT | 78.7 | 70.6 | 63.8 | **72.9** | 54.1 | **76.3** |

Table 3: Results on the Tweet-Stance dataset. The $\text{F1}_{avg}$ column represents the F1 score on the full test set. Per-topic F1 score is additionally reported above by sub-setting TweetStance by topic and re-computing the F1 score. Results marked with † are taken from prior work.

(131/175) of the samples are on tweets incorrectly labeled as neutral by ChatGPT COT. This confirms our intuition that passing COT information to text encoders may help solve the *Implicit Stance Confusion* problem. Of the remaining 44 samples TR-Tweet+COT was able to predict correctly, we manually inspected the 20/44 where ChatGPT predicts "Against" when the true label was "In Favor". We find that 9/9 samples from the HC, FM, LA, AT topics are examples of *stance label hallucination*. For example, consider the COT reasoning: " . . . it is clear that [NO] this text is against Jeb Bush and in favor of Hillary". This text was marked "[NO] = Against Hillary" by our COT parser but was able to be handled by our encoder model as the reasoning was accurate. The remaining 11 samples in this analysis are from the climate change topic, where most COT errors largely pertain to questions of what it means to be "in favor" or "against" climate change, which we view as more of a natural misunderstanding than instances of stance label hallucination. Future works may explore better prompts to elicit better predictions on climate change tweets.

In Table 5, we evaluate the performance of COT produced by different LLMs. We find that while ChatGPT produces the highest performing COT, we achieve a meaningful performance increase when employing the smaller open-source LLM Llama-2-7b[3] (Touvron et al., 2023). Unfortunately,

lower-performing LLMs such as Falcon-7b [4] (Almazrouei et al., 2023) do not provide useful COT, highlighting the importance of LLM performance on this task.

## 4.2 Presidential-Stance

Table 4 presents the results of the Presidential-Stance dataset. Results indicate that our approach outperforms all baseline models. When we analyze the Biden data, TR-Tweet+COT **outperforms previous works by 1.4 F1-pts**. A very interesting result is the extreme difference in performance between ChatGPT-COT and TR-COT, which provides a 20.7-pt boost in F1 score. This is driven by a large number of *Implicit Stance Confusion* examples where it's challenging to understand the label without seeing other training samples. Specifically, our model is correcting Neutral class predictions 56% of the time — as ChatGPT can assume mentions of democratic figures or ideals are taking a stance on Joe Biden — which is not always the case, causing under-prediction on Neutral samples. Our error analysis also found stance label hallucinations as ChatGPT was found to go off-topic when the focus of the tweet is on another political figure: "wow bernie sander is the only one who supports democracy #demdebate" provoked a ChatGPT response of "... this tweet is [IN FAVOR] of Bernie Sanders." which is of course not the question being asked.

Similarly, on the Trump data, we find that our best-performing model **outperforms the closest baseline by 2.4 F1-pts**. Interestingly, we note that our best model *does not use the tweet information at all*, as TR-COT obtains the highest average F1 score (81.5). This outcome suggests that the COT reasoning is often logically sound, but our TR-COT model makes the predictions more robust to errors in the ChatGPT COT output structure.

In Table 5, we again evaluate the performance of COT produced by different LLMs on Presidential Stance. We find that on both the Biden and Trump datasets, ChatGPT provides the highest performing COT. On both the Biden and Trump datasets, we also find that Llama-2 performs much better than Falcon, again highlighting the importance of LLM quality in our pipeline. Notably, Llama-2 only provides helpful COT for the Biden dataset, not Trump. This result, however, is expected as ChatGPT, a higher-performing language model than Llama-2-

---

[3]https://huggingface.co/meta-llama/Llama-2-7b-chat-hf

[4]https://huggingface.co/tiiuae/falcon-7b-instruct

|  | Biden | | | | Trump | | | |
|---|---|---|---|---|---|---|---|---|
| Model | F | A | N | Avg | F | A | N | Avg |
| *Baselines* | | | | | | | | |
| BERT[†] | 73.2 | 68.7 | 71.5 | 71.1 | 75.7 | 81.0 | 69.5 | 75.4 |
| SKEP[†] | 79.2 | 71.5 | 73.4 | 74.7 | 78.5 | 81.6 | 71.5 | 77.2 |
| KE-MLM[†] | 79.2 | 73.2 | 74.7 | 75.7 | 80.9 | 81.8 | 73.5 | 78.7 |
| *ChatGPT Only* | | | | | | | | |
| 0-shot | 62.1 | 57.9 | 65.9 | 62.0 | 66.3 | 69.4 | 65.8 | 67.1 |
| COT | 55.0 | 52.7 | 44.2 | 50.6 | 77.0 | 79.9 | 71.6 | 76.2 |
| *COT-Embeddings + Twitter-RoBERTa* | | | | | | | | |
| TR-Tweet | 77.4 | 72.6 | 71.8 | 73.9 | 81.6 | 82.6 | 73.0 | 79.1 |
| TR-COT | 74.2 | 71.5 | 68.2 | 71.3 | 81.6 | **85.5** | **77.6** | **81.5** |
| TR-Tweet + COT | **80.6** | **75.9** | **74.8** | **77.1** | **82.3** | 84.8 | 75.1 | 80.7 |

Table 4: F1 scores for both the Biden and Trump subsets of the Presidential-Stance dataset. We show class-wise performance on Favor, Against, and Neutral classes. The average score is the Macro-F1 score across each class. All *COT-Embeddings + Twitter-RoBERTa* experiments are the average score of three experimental trials. Baseline results marked with † are taken from prior work. Standard deviation of each experiment is shown in Appendix D.

| Model | Tweet Stance | P-Biden | P-Trump |
|---|---|---|---|
| *Baselines* | | | |
| TR-Tweet | 69.0 | 73.9 (0.87) | 79.1 (0.59) |
| *Falcon-7b-instruct* | | | |
| TR-COT | 66.5 | 53.0 (1.60) | 61.9 (1.11) |
| TR-Tweet+COT | 65.3 | 76.0 (0.99) | 76.8 (0.19) |
| *Llama-2-7b-chat* | | | |
| TR-COT | 69.3 | 63.9 (0.49) | 69.1 (1.08) |
| TR-Tweet+COT | 72.5 | 77.0 (0.47) | 78.4 (0.73) |
| *ChatGPT (gpt-3.5)* | | | |
| TR-COT | 75.7 | 71.3 (1.28) | **81.5 (0.33)** |
| TR-Tweet+COT | **76.3** | **77.1 (1.46)** | 80.7 (0.61) |

Table 5: Comparing the F1 score of different LLMs on Tweet Stance, Presidential Stance Biden (P-Biden) and Presidential Stance Trump (P-Trump). Recall that Presidential Stance has no development set, thus we report the mean result (and standard deviation) over three experimental trials. Our results find that, in general ChatGPT is the highest performing LLM. We also validate our approach works using Llama-2, a smaller open-source model.

7b, only provides a minor improvement over the baseline TR-Tweet.

# 5 Conclusion

In this study, we have shown that embedding Chain-of-Thought reasoning extracted from LLMs (e.g., ChatGPT, Lllama) can boost the performance of stance detection models. Specifically, we highlight how we can outperform vanilla COT by augmenting text encoders with COT embedding. Our analysis highlights how text encoders are robust to LLM hallucinations and aid in the prediction of deeply implicit stance labels. We encourage future works to consider embedding COT reasoning for stance detection and similar tasks using social media data.

# 6 Limitations

A limitation of this work is that stance detection using COT reasoning is very sensitive to the prompt provided to ChatGPT (Zhang et al., 2023b). In this study, we do not thoroughly investigate which COT prompt produces the best results, but rather try a few standard approaches inspired by related works. Future works aiming to optimize COT prompt structure for stance detection may find ways to reduce the effects of error hallucinations. In general, our work reduces the need for prompt optimization by mitigating issues pertaining to common COT errors.

Another limitation of this work is that one of its core takeaways — that COT Embeddings reduce effects of implicit stance confusion — may only be applicable to popular social media platforms where colloquial language is constantly changing. Application of COT Embeddings to other domains where all necessary information for inference is present in a single sample (e.g., in certain NLI tasks), COT Embeddings may not be as helpful.

Finally, we note that the addition of COT embeddings may impact the computational efficiency of the model. Specific measures of computational efficiency are currently outside the scope of this paper. However, we highlight that if one is in a setting where the COT reasoning can be pre-computed, the impact of COT on computational efficiency is limited. While if COT reasonings had to be computed at inference time, there may be noticeable inference speed degradation depending on the efficiency of the LLM used for COT reasoning.

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

# Appendix

## A  Training Details

**Tweet-Stance:**  For training, we aggregate the training set of all stance topics. Since there are a mix of topics in the training set, we include the topic as part of the input (i.e. pretend tweet with topic information) during training. We train each model with a batch size of 16 for a maximum of 10 epochs with early stopping on the development set with patience = 2 keeping other related parameters default [5]. The learning rate for each of our models was chosen via a modest grid search of [1e-3, 1e-4, 2e-5, 5e-5]. All other parameters are the default provided by the Huggingface Trainer [6].

**Presidential-Stance:**  For training, since there is no available development set, we simply fine-tune each model with a batch size of 32 for 5 epochs using the default parameters provided by the Huggingface trainer. We report the average macro F1 score and report the standard deviation over three experimental runs with different random seeds.

## B  Prompting Details

In our experiments, we use the following COT prompting template.

> Read the following tweet and decide if the stance of the tweet is in favor [IN FAVOR / YES], against [AGAINST / NO], or neutral [NEUTRAL / NONE] with regards to the topic <topic>:
>
> Tweet: <Tweet>
> Stance: <COT Example>
>
> Tweet: <Tweet>
> Stance: Lets think about this step by step.

The label to be output by the model is in brackets [] and is used to parse the output string to convert the label into a numeric representation. Notice that we explore two label structures, the supporting label is either [IN FAVOR] or [YES], the non-supporting label is either [AGAINST] or [NO], and the stance-free label is either [NEUTRAL] or [NONE]. We provide examples of both label

[5]https://huggingface.co/docs/transformers/main_classes/callback#transformers.EarlyStoppingCallback
[6]huggingface.co/docs/transformers/main_classes/trainer

structures in the samples below. The <Tweet> and <Topic> markers are parameters of the prompt and dynamically change given the sample. Consider the following two examples which COT prompt for stances on Atheism and Donald Trump:

> **Example 1:** Read the following tweet and decide if the stance of the tweet is in favor [YES], against [NO], or neutral [NONE] with regards to the topic Atheism:
>
> Tweet: You cant think by yourself about life and believe in god. It just doesn't add up #SemST
> Stance: Lets think step by step. Since this text finds belief in god to be contradicting with the notion of thinking by oneself, it must be the case that [YES] this text is in favor of atheism.
>
> Tweet: <Tweet>
> Stance: Lets think about this step by step.

> **Example 2:** Read the following tweet and decide if the stance of the tweet is in favor [IN FAVOR], against [AGAINST], or neutral [NEUTRAL] with regards to the topic Joe Biden:
>
> Tweet: america's ceos say trump failed on coronavirus – and they're backing biden HTTP
> Stance: Lets think step by step. Since the tweet mentions Trump's failures and how important CEOs in america are backing Joe Biden, then this tweet is [IN FAVOR] of Joe Biden.
>
> Tweet: <Tweet>
> Stance: Lets think about this step by step.

1-shot COT examples were chosen randomly from each training set with the COT reasoning written by the author. We note that our 0-Shot ChatGPT baseline uses the same prompt without the 1-shot COT example.

## C Variability of Presidential-Stance Results

Due to space constraints we are unavailable to add the standard deviations of our Presidential-Stance experiments to Table 4 in the main paper. Please refer to Table 6 for the standard deviation of each experiment.

| Model | Biden | Trump |
|---|---|---|
| **TR-Tweet** | $73.9 \pm 0.87$ | $79.1 \pm 0.59$ |
| **TR-COT** | $71.3 \pm 1.28$ | $81.5 \pm 0.33$ |
| **TR-Tweet+COT** | $77.1 \pm 1.46$ | $80.7 \pm 0.61$ |

Table 6: Macro F1 scores with standard deviations over three experimental trails with different random seeds.

## D Variability of Tweet-Stance Results

While Tweet-Stance results are the product of dev set optimization, we also re-run our model five times with five different random seeds to highlight model variability, as this is a fairly low-resource problem. Please refer to Table 7 for the resulting standard deviations of each experiment.

| Model | F1 |
|---|---|
| **TR-Tweet** | $70.58 \pm 1.20$ |
| **TR-COT** | $76.53 \pm 0.54$ |
| **TR-Tweet+COT** | $76.55 \pm 1.72$ |

Table 7: F1 scores on Tweet-Stance with standard deviations over five experimental trails with different random seeds.