# OpenReview forum: "Chain-of-Thought Embeddings for Stance Detection on Social Media"
_EMNLP/2023/Conference — EMNLP 2023 Findings_

### Official Review · Reviewer_XojB · 2023-07-24

**Soundness:** 4

**Excitement:**

4: Strong: This paper deepens the understanding of some phenomenon or lowers the barriers to an existing research direction.

**Paper Topic And Main Contributions:**

This short paper is about stance detection on social media posts, where stance is often difficult to detect. The paper proposes to use a large language model, while not extracting the label directly from the LM output. Instead, a BERT classifier is trained to use the LM output as an additional (or alternative) feature.
Experiments show an improvement over more simple baselines on political stance detection of English tweets.

**Questions For The Authors:**

A. How do COT Embeddings affect the time needed for inference, compared to direct classification?

B. Appendix C: Consider reporting these numbers for the Tweet-Stance dataset as well.

**Reasons To Accept:**

- The two-step approach presented in this paper is a simple, and likely novel, idea that is relevant for a range of text classification tasks.

**Reasons To Reject:**

- Limited reproducibility: ChatGPT, a commercial product, was used as the language model and it might not be possible for future work to reproduce the results. The authors do not indicate that they will share the LM outputs they collected.

**Reproducibility:**

5: Could easily reproduce the results.

**Reviewer Confidence:**

3: Pretty sure, but there's a chance I missed something. Although I have a good feel for this area in general, I did not carefully check the paper's details, e.g., the math, experimental design, or novelty.

---

> ### Author Rebuttal · Authors · 2023-08-28
>
> ## Overview
>
> We kindly thank the reviewer for their feedback on our paper. We are encouraged by the reviewers’ high excitement score, as well as the reviewers appreciation of our intuitive and  novel approach to applying Chain-of-Thought (COT) embeddings to stance detection. We specifically appreciate the reviewer taking note of our methods' potential applicability to other downstream text classification tasks. Please find our specific response to the reviewer’s comments and questions below.
>
> ## Questions/Comments
>
> ___
> > **Comment (1):** “Limited reproducibility: ChatGPT, a commercial product, was used as the language model and it might not be possible
> for future work to reproduce the results.”
>
> **Response (1):** We agree with the reviewer that our results will be more reproducible if we can validate that the embeddings of COT-reasonings from other open-source LLMs can improve baseline performance. We have thus added two additional experiments using the popular Llama 2 [1] and [Falcon](https://huggingface.co/tiiuae/falcon-7b-instruct) LLMs. The results of the additional experiments can be found in Table 1 and Table 2. Recall that we use the Twitter-RoBERTa (TR) backbone to encode the tweet-only (TR-Tweet), the COT only (TR-COT), and both the Tweet and COT texts together (TR-Tweet+COT). Note that gpt-3.5-turbo results are from the original paper. LLama-2 and Falcon results are additional results run as part of the rebuttal.
>
> Our experiments confirm that with the recently released [llama-2-7b-chat](https://huggingface.co/meta-llama/Llama-2-7b-chat) our results hold for most experiments. Specifically, embedding Llama 2 COT reasonings gives us a 3.1 pt boost on Presidential-Stance (Biden Subset) and a 3.5 pt boost on Tweet-Stance . However, we find that results do not improve with Llama 2 COT embeddings on the Trump subset of the Presidential-Stance  dataset. This however was expected as gpt-3.5-turbo, a model much more powerful than Llama2-7b-chat (as per results in [1]) only provides a minor improvement over the baseline TR-Tweet (Trump).
>
> We find that COT-reasonings from [falcon-7b-instruct](https://huggingface.co/tiiuae/falcon-7b-instruct) less impactful, however this is expected as Falcon is significantly less powerful than Llama-2 (as per [1]). Thus, our additional experiments confirm that *LLM performance is critical to the effectiveness of the proposed CoT based approach.* Please see the tables below for the full details of these additional experiments.
>
> |                | LLMs used to generate COT reasoning |  Biden      | Trump       |
> |----------------|--------------------------------------|-------------|-------------|
> | TR-Tweet       | -                                    | 73.9 (0.87) | 79.1 (0.59) |
> |   TR-COT       |             gpt-3.5-turbo            | 71.3 (1.28) | **81.5 (0.33)** |
> |                |            Llama-2-7b-chat           | 63.9 (0.49) | 69.1 (1.08) |
> |                |          Falcon-7b-instruct          | 53.0 (1.60) | 61.9 (1.11) |
> |   TR-Tweet+COT |             gpt-3.5-turbo            | **77.1 (1.46)** | 80.7 (0.61) |
> |                |            Llama-2-7b-chat           | 77.0 (0.47) | 78.4 (0.73) |
> |                |          Falcon-7b-instruct          | 76.0 (0.99) | 76.8 (0.19) |
>
> **Table 1:** F1 scores on Presidential-Stance using COT embeddings from 3 different LLMs. Due to the lack of development set in Presidential Stance,  results are reported as the mean of three experimental trials. The standard deviation is reported in parentheses.
>
>
> |                | LLMs used to generate COT reasoning | F1-score |
> |----------------|--------------------------------------|----------|
> | TR-Tweet       |                                      |   69.0   |
> |   TR-COT       |             gpt-3.5-turbo            | **75.7** |
> |                |            Llama-2-7b-chat           |   69.3   |
> |                |          Falcon-7b-instruct          |   66.5   |
> |   TR-Tweet+COT |             gpt-3.5-turbo            | **76.3** |
> |                |            Llama-2-7b-chat           |   72.5   |
> |                |          Falcon-7b-instruct          |   65.3   |
>
> **Table 2.**  F1 scores on Tweet-Stance using COT embeddings on 3 different LLMs. Unlike Presidential-Stance, Tweet-Stance has a development set, so we report results after early stopping as mentioned in Appendix A.
>
> We will share these new results in the main paper (as space permits) and/or in the appendix.
>
> ___
> > **Comment (2):** “The authors do not indicate that they will share the LM outputs they collected.”
>
> **Response (2):** Thank you for this great idea! We will be sure to release the LLM outputs along with our code upon acceptance of the paper!
>
> ___
> > **Question (A):** How do COT Embeddings affect the time needed for inference, compared to direct classification?
>
> **Response (A):** This is an important question that should be addressed in detail in future works — however measures of computational efficiency are currently outside the scope of this paper. Intuitively if one is in a setting where the COT-reasoning can be pre-computed, the impact is limited. However, if COT-reasonings had to be computed at inference time, there would be noticeable inference speed degradation. Hopefully this answers your question!
>
> ___
> > **Question (B):** “Appendix C [Referring to standard deviation of Presidential-Stance results]: Consider reporting these numbers for the Tweet-Stance dataset as well.”
>
> **Response (B):** We can certainly report the mean result over numerous trials on Tweet-Stance in the final version of our paper. However, we remind the reviewer that our Tweet-Stance results were optimized using the development set and that Presidential-Stance does not have a development set, which provoked the need for reporting mean performance. However, given that there may be variability in results with different seeds, we have run this experiment on Tweet-Stance and report the new results below.
>
> |  Experiment  |   F1  | Standard Deviation |
> |:------------:|:-----:|:------------------:|
> |   TR-Tweet   | 69.8  |        1.27        |
> |    TR-COT    |  76.0 |        1.10        |
> | TR-Tweet+COT |  77.6 |      1.11e-16      |
>
> **Table 3:**  Mean and standard deviation of Tweet-Stance with ChatGPT COT Embeddings after 3 experimental trials with different random seeds. Results using COT leverage outputs from ChatGPT (gpt-3.5-turbo).
>
> We will share these new results in the appendix or our final manuscript.
>
> ---
>
> We appreciate your time and efforts to review our paper! We hope this response satisfies your concerns and resolves any potential ambiguity. Please let us know if our replies address your concerns.
>
> ---
> ### References
> ___
> [1] Touvron, H., Martin, L., Stone, K., Albert, P., Almahairi, A., Babaei, Y., ... & Scialom, T. (2023). Llama 2: Open foundation and fine-tuned chat models. arXiv preprint arXiv:2307.09288.

---

### Official Review · Reviewer_pvkN · 2023-08-03

**Soundness:** 3

**Excitement:**

3: Ambivalent: It has merits (e.g., it reports state-of-the-art results, the idea is nice), but there are key weaknesses (e.g., it describes incremental work), and it can significantly benefit from another round of revision. However, I won't object to accepting it if my co-reviewers champion it.

**Paper Topic And Main Contributions:**

The paper presents an approach for stance detection in social media, using Chain-of-Thought (COT) embeddings along with texts to be classified.
This approach follows (Zhang et al, 2023) and asks ChatGPT to generate reasoning texts (COT) for a given tweet and to predict its stance towards a given topic. Then, COT are used as a feature in a stance classification model (RoBERTa) in order to inject world knowledge instead of using a real knowledge base.
This model is evaluated on 2 datasets and outperforms other baselines, including ChatGPT.


**Questions For The Authors:**

A) line 79: "COT reasoning is an inexpensive way to inject world knowledge into
a text encoder"
=> I agree with this statement but I would not say that using ChatGPT (here to generate COT) is "inexpensive"

B) The datasets used for evaluation are rather small (for example, 1800 intances for the class FAVOR). Table 1 should also give the distribution for each class in the train/test sets.

C) Why not testing all baseline models on both datasets?

D) line 155: "tweet and COT reasoning are treated as a pair-wise input to the model"
=> What do you mean by "pair-wise input"? Are the embeddings concatenated? or do you mean a pair-wise learning method which attempts to find a loss function for feature pairs?

E) line 202: "Results show that TR-Tweet+COT produces the best-performing model on Tweet-Stance"
=> However when looking per topic, TR-COT seems to be better than TR-Tweet+COT.
=> In Table 2 caption, I don't understand "Per-class F1 score is additionally reported": F1-score are given per topic, but not per class.

F) line 471: "1-shot COT examples were chosen randomly from each training set with the COT reasoning written by the author. We note that our 0-Shot ChatGPT baseline uses the same prompt without the 1-shot COT example."
=> How many examples were chosen for the 1-shot COT?
=> In example 2 (line 453): the topic is Trump but the 1-shot COT example is about Biden (which is called label hallucination in the paper). Has the author written an incorrect example on purpose?


**Reasons To Accept:**

- Embedding COT reasoning texts in a stance classification model improves the results.


**Reasons To Reject:**

- The datasets used for evaluation are rather small.
- Some details are unclear (prompting, distribution, F1-score...)

**Reproducibility:**

3: Could reproduce the results with some difficulty. The settings of parameters are underspecified or subjectively determined; the training/evaluation data are not widely available.

**Reviewer Confidence:**

4: Quite sure. I tried to check the important points carefully. It's unlikely, though conceivable, that I missed something that should affect my ratings.

---

> ### Author Rebuttal · Authors · 2023-08-28
>
> ## Overview
> We thank the reviewers for their very thoughtful and detailed comments on our paper. We believe this feedback will make our work stronger. We are encouraged by the reviewers’ appreciation of our Chain-of-Thought (COT) embedding approach and how it improves results over previous works. We appreciate the questions and feedback the reviewer provided regarding dataset size and evaluation strategy. We address all reviewer comments and questions below.
>
> ## Questions
>
> ----
> > **Question (A):** “line 79: "COT reasoning is an inexpensive way to inject world knowledge into a text encoder" => I agree with this statement but I would not say that using ChatGPT (here to generate COT) is "inexpensive"”
>
> **Response (A):**  This is a great point and perhaps a poor choice of words, as the use of OpenAI’s ChatGPT API is not free to use. In this line, we are trying to communicate that it is low-cost with respect to the lack of need for additional model training parameters + lower implementation difficulty. We will be sure our final draft reflects this change in phrasing.
>
> ---
> > **Question (B.1):** “The datasets used for evaluation are rather small (for example, 1800 instances for the class FAVOR).
>
> **Response (B.1):** We agree with the reviewer but note that we are unfortunately restricted to evaluation on available public datasets for which there are limited available resources. Stance-detection on social media is often inherently resource constrained due to significant expense and effort of large scale annotation. In the future we hope to evaluate our work on larger datasets. Note that related works from top conferences such as [1,2,3] also utilize Tweet-Stance for evaluation.
>
> ---
> > **Question (B.2):** “Table 1 should also give the distribution for each class in the train/test sets.”
>
> **Response (B.2):** Thank you for this great suggestion. Please find this information below for reference.
>
> | Split | Favor | Against | Neutral |
> |:-----:|:-----:|:-------:|:-------:|
> | Train | 678   | 1254    | 688     |
> | Dev   | 75    | 141     | 78      |
> | Test  | 304   | 715     | 230     |
>
> **Table 1:** Class distribution for each data split on Tweet-Stance
>
> |    Split    | Favor | Against | Neutral |
> |:-----------:|:-----:|:-------:|:-------:|
> | Biden-Train | 266   | 279     | 330     |
> | Biden-Test  | 112   | 106     | 157     |
> | Trump-Train | 243   | 347     | 285     |
> | Trump-Test  | 98    | 152     | 125     |
>
> **Table 2:** Class distribution for each data split on Presidential-Stance
>
> If space permits, we will add this information to the main paper, otherwise we will be sure to include it in the appendix.
>
> ____
> > **Question (C):**  “Why not testing all baseline models on both datasets?”
>
> **Response (C):** We did not run all baselines ourselves. Rather, certain baseline results were listed as they had been reported in prior works, where evaluation on certain stance topics was missing. We are unable to fill-in missing baselines from prior works by the rebuttal deadline but will consider doing so for our final submission. This issue arises as some results reported from prior works unfortunately only cite the model used for the baseline without providing training details. We would thus be unable to run the experiment ourselves in a consistent manner.
>
> ___
> > **Question (D):**  line 155: "tweet and COT reasoning are treated as a pair-wise input to the model" => What do you mean by "pair-wise input"? Are the embeddings concatenated? or do you mean a pair-wise learning method which attempts to find a loss function for feature pairs?
>
> **Response (D):**  We refer to pairwise input in the sense of how one might pass a pairwise task to a BERT-based model for text classification. (See Figure 2 in the [BERT](https://arxiv.org/abs/1810.04805) paper for a visualization). Specifically, we feed <s> Tweet </s> COT-Reasoning </s> to the RoBERTa model. Thus, the Tweets and COT reasoning are cross-encoded together in this manner. We will clarify this point in the final draft of our manuscript.
>
> ___
> > **Question (E.1):** “line 202: "Results show that TR-Tweet+COT produces the best-performing model on Tweet-Stance" => However when looking per topic, TR-COT seems to be better than TR-Tweet+COT.
>
> **Response (E.1):** This is correct. However, our conclusion that TR-Tweet+COT is the best-performing model is based on the overall F1 score, which is performance on the full test set. We include per-topic scores for reference, but given that different topics have different distributions / different numbers of samples, we think that it’s unfair to classify a model as higher performing because it got a higher score on certain topics.
>
> For example, Tweet-Stance  F1 is defined as mean F1 on **Favor** and **Against**, with no consideration of **Neutral** predictions. The Climate Change topic only has 26 **Against** samples vs the Hillary Clinton topic which has 565 **Against** samples.
>
> ---
> > **Question (E.2):** “In Table 2 caption, I don't understand "Per-class F1 score is additionally reported": F1-score are given per topic, but not per class.”
>
> **Response (E.2):**  We apologize for the confusion on this point. You are correct in stating that the terminology we should have used is “Topic” not “class”, as class would refer to (Favor, Against, Neutral) for this task. Here we are showing the overall F1 score as well as per-topic F1 score. This typo will be fixed in our final manuscript.
>
> ___
> > **Question (F.1):**  “line 471: "1-shot COT examples were chosen randomly from each training set with the COT reasoning written by the author. We note that our 0-Shot ChatGPT baseline uses the same prompt without the 1-shot COT example." => How many examples were chosen for the 1-shot COT?”
>
> **Response(F.1):** We randomly select one sample per-topic in each training set. For example, since there are 5 topics in Tweet-Stance , there are 5 total samples used for 1-shot COT prompting. When we pass a Hillary Clinton sample to an LLM for 1-shot COT, it would have a different demonstration than when we pass a Climate Change sample. However, only one demonstration is shown in both cases. We will clarify this point in our final manuscript!
>
> Note that we did no prompt optimization in this study as it was considered out of the scope of this work. We simply pick a random sample and write one correct COT reasoning to help guide the model on the task.
>
> ____
> > **Question (F.2):** “In example 2 (line 453): the topic is Trump but the 1-shot COT example is about Biden (which is called label hallucination in the paper). Has the author written an incorrect example on purpose?”
>
> **Response (F.2):** We apologize for this manuscript typo as the underlined topic in this example should read “Joe Biden”. This will be fixed in the final version of the manuscript.
>
> ---
>
> We appreciate your time and efforts to review our paper! We hope this response satisfies your concerns and resolves any potential ambiguity. Please let us know if our replies address your concerns.
>
> ---
>
> ### References
> ___
> [1] Yang Li and Jiawei Yuan. 2022. Generative Data Augmentation with Contrastive Learning for Zero-Shot Stance Detection. In Proceedings of the 2022 Conference on Empirical Methods in Natural Language Processing, pages 6985–6995, Abu Dhabi, United Arab Emirates. Association for Computational Linguistics.\
> [2] Yingjie Li and Cornelia Caragea. 2021. Target-Aware Data Augmentation for Stance Detection. In Proceedings of the 2021 Conference of the North American Chapter of the Association for Computational Linguistics: Human Language Technologies, pages 1850–1860, Online. Association for Computational Linguistics.\
> [3] Matero, M., Soni, N., Balasubramanian, N., & Schwartz, H. A. (2021). MeLT: Message-level transformer with masked document representations as pre-training for stance detection. ACL Findings

---

### Official Review · Reviewer_bcgD · 2023-08-11

**Soundness:** 2

**Excitement:**

3: Ambivalent: It has merits (e.g., it reports state-of-the-art results, the idea is nice), but there are key weaknesses (e.g., it describes incremental work), and it can significantly benefit from another round of revision. However, I won't object to accepting it if my co-reviewers champion it.

**Paper Topic And Main Contributions:**

This manuscript is focused on the issue of Chain-of-Thought for stance detection. This paper proposes the CoT embedding method and the model is evaluated on several stance detection dataset, and is shown to outperform a variety of baseline models.

Strengths: This work tackles a useful task, and the general approach seems novel and interesting to many researchers.

Weaknesses:
The experimental comparison is limited, despite the availability of more recent models published between 2021 and 2023.
The contribution of the paper seems incremental - applying some variation of known methods to a known problem.
The writing could be improved.

**Reasons To Accept:**

I think that the paper would be of interest for people working on stance classification. This work tackles a useful task and the approach is reasonable.

**Reasons To Reject:**

As stated above, the paper makes incremental contributions to existing problems, therefore, I think its impact is limited.

**Reproducibility:**

3: Could reproduce the results with some difficulty. The settings of parameters are underspecified or subjectively determined; the training/evaluation data are not widely available.

**Reviewer Confidence:**

4: Quite sure. I tried to check the important points carefully. It's unlikely, though conceivable, that I missed something that should affect my ratings.

---

> ### Author Rebuttal · Authors · 2023-08-28
>
> ## Overview
>
> We thank the reviewers for their comments on our paper. We are encouraged by the reviewer's view that our work on embedding Chain-of-Thought (COT) outputs from LLMs is novel, useful to the task of stance detection, and interesting to the research community. We also appreciate the feedback regarding the quality of our experimental comparison. We hope to address all of the reviewers' questions and concerns below.
>
> ## Questions
> ---
> > **Question (Q1):** “The experimental comparison is limited, despite the availability of more recent models published between 2021 and 2023.”
>
> **Answer (1):** Assuming that this comment is in reference to the use of only one LLM, i.e., ChatGPT, which has a knowledge cutoff of 2021, we address this concern via additional experiments on recently released LLMs. Specifically, we evaluate our approach using both Falcon and Llama 2 [1], which has knowledge through September 2022. The results of the additional experiments can be found in Table 1 and Table 2. Recall that we use the Twitter-RoBERTa (TR) backbone to encode the tweet-only (TR-Tweet), the COT only (TR-COT), and both the Tweet and COT texts together (TR-Tweet+COT). Note that gpt-3.5-turbo results are from the original paper. LLama-2 and Falcon results are additional results run as part of the rebuttal.
>
> Our experiments confirm that with the recently released [llama-2-7b-chat](https://huggingface.co/meta-llama/Llama-2-7b-chat) our results hold for most experiments. Specifically, embedding Llama 2 COT reasonings gives us a 3.1 pt boost on Presidential-Stance (Biden Subset) and a 3.5 pt boost on Tweet-Stance . However, we find that results do not improve with Llama2 COT embeddings on the Presidential-Stance dataset (Trump Subset). This however was expected as gpt-3.5-turbo, a model much more powerful than Llama2-7b [1], only provides a minor improvement over the baseline TR-Tweet (Trump). We find that COT-reasonings from [falcon-7b-instruct](https://huggingface.co/tiiuae/falcon-7b-instruct) are less impactful, however this is expected as Falcon is significantly less powerful than Llama-2 (as indicated in [1]). Thus, *our additional experiments confirm that LLM performance is critical to the effectiveness of the proposed COT based approach.* Please see the table below for the full details of these additional experiments.
>
> |                | LLMs used to generate COT reasoning |  Biden      | Trump       |
> |----------------|--------------------------------------|-------------|-------------|
> | TR-Tweet       | -                                    | 73.9 (0.87) | 79.1 (0.59) |
> |   TR-COT       |             gpt-3.5-turbo            | 71.3 (1.28) | **81.5 (0.33)** |
> |                |            Llama-2-7b-chat           | 63.9 (0.49) | 69.1 (1.08) |
> |                |          Falcon-7b-instruct          | 53.0 (1.60) | 61.9 (1.11) |
> |   TR-Tweet+COT |             gpt-3.5-turbo            | **77.1 (1.46)** | 80.7 (0.61) |
> |                |            Llama-2-7b-chat           | 77.0 (0.47) | 78.4 (0.73) |
> |                |          Falcon-7b-instruct          | 76.0 (0.99) | 76.8 (0.19) |
>
> **Table 1:** F1 scores on Presidential-Stance using COT embeddings from 3 different LLMs. Due to the lack of development set in Presidential Stance,  results are reported as the mean of three experimental trials. The standard deviation is reported in parentheses.
>
>
> |                | LLMs used to generate COT reasoning | F1-score |
> |----------------|--------------------------------------|----------|
> | TR-Tweet       |                                      |   69.0   |
> |   TR-COT       |             gpt-3.5-turbo            | **75.7** |
> |                |            Llama-2-7b-chat           |   69.3   |
> |                |          Falcon-7b-instruct          |   66.5   |
> |   TR-Tweet+COT |             gpt-3.5-turbo            | **76.3** |
> |                |            Llama-2-7b-chat           |   72.5   |
> |                |          Falcon-7b-instruct          |   65.3   |
>
> **Table 2.**  F1 scores on Tweet-Stance using COT embeddings on 3 different LLMs. Unlike Presidential-Stance, Tweet-Stance has a development set, so we report results after early stopping as mentioned in Appendix A.
>
> We will share these new results in the main paper (as space permits) and in the appendix.
>
> ----
>
> > **Question (Q2):** “The contribution of the paper seems incremental - applying some variation of known methods to a known problem...I think its impact is limited.”
>
> **Answer (A2):** We appreciate your perspective on this matter. However, we might have different interpretations. We want to share additional details that might clarify our point. Our work does not necessarily apply a *variation* of known methods but rather a *novel* combination of known methods. We believe this is a relevant and timely contribution for the following reasons.
> Our technical implementation is simple and intuitive for use in future downstream NLP applications. We provide novelty as we are the first to embed COT reasonings as a means for providing additional context and world knowledge to the stance detection pipeline, unlike prior works that simply use COT for text classification. Additionally, we believe advancements in stance detection are of high impact as it is a task of great importance for large-scale aggregation of public opinion. Our work provides an increase in performance over prior works at a rate consistent with prior publications on Stance Detection @ EMNLP such as [2,3].
>
> ----
> We appreciate your time and efforts to review our paper! We hope this response satisfies your concerns and resolves any potential ambiguity. Please let us know if our replies address your concerns.
>
> ----
> ### References
>
> ---
> [1] Touvron, H., Martin, L., Stone, K., Albert, P., Almahairi, A., Babaei, Y., ... & Scialom, T. (2023). Llama 2: Open foundation and fine-tuned chat models. arXiv preprint arXiv:2307.09288.\
> [2] Yingjie Li, Chenye Zhao, and Cornelia Caragea., Improving Stance Detection with Multi-Dataset Learning and Knowledge Distillation. (EMNLP 2021)\
> [3] Matero, M., Soni, N., Balasubramanian, N., & Schwartz, H. A., MeLT: Message-level transformer with masked document representations as pre-training for stance detection. (EMNLP Findings 2021)

---

### Official Review · Reviewer_n7UD · 2023-08-11

**Soundness:** 3

**Excitement:**

3: Ambivalent: It has merits (e.g., it reports state-of-the-art results, the idea is nice), but there are key weaknesses (e.g., it describes incremental work), and it can significantly benefit from another round of revision. However, I won't object to accepting it if my co-reviewers champion it.

**Paper Topic And Main Contributions:**

This paper explores to incorporate the embedding of chain-of-thought reasoning generated by chatgpt into traditional pre-trained Twitter-roberta fine-tuning process.

**Questions For The Authors:**

For the 1-shot COT proposed by the authors, how sensitive the approach is to the COT reasoning written by the authors?

**Reasons To Accept:**

The topic and approaches are relatively novel for stance detection.

**Reasons To Reject:**

I have following concerns:
1) In-depth analysis and more datasets needed. I suspect the improvement of the performance may be due to the existence of certain stance-related cue words generated by chatgpt, rather than injecting world knowledge. The paper could be more informative if the authors could provide some analysis on the generated COT reasonings. I'm also not convinced that COT embedding could help implicit stance detection, since the authors only provide related analysis on one dataset (Twitter Stance).
2) Data leakage. it is possible that chatgpt has already seen these stance datasets during its training. It may be interesting to compare the results on other LLMs with chatgpt.

**Reproducibility:**

3: Could reproduce the results with some difficulty. The settings of parameters are underspecified or subjectively determined; the training/evaluation data are not widely available.

**Reviewer Confidence:**

3: Pretty sure, but there's a chance I missed something. Although I have a good feel for this area in general, I did not carefully check the paper's details, e.g., the math, experimental design, or novelty.

---

> ### Author Rebuttal · Authors · 2023-08-28
>
> ### Overview
> We thank the reviewers for their thoughtful feedback. We are encouraged that the reviewer finds our method of embedding Chain-of-Thought (COT) reasonings from Large Language Models (LLMs) to be novel. We specifically appreciate the reviewer’s feedback on our analysis, number of evaluation datasets, and for highlighting the potential of data leakage via the use of ChatGPT. We answer specific questions and feedback from the reviewer below.
>
>
> ###  Questions
> ___
> > **Question (Q1.1):**  “More datasets needed.“
>
> **Answer (A1.1):** We highlight that our choice of evaluating on two benchmark datasets is in-line with prior works on stance detection which evaluate on 1-3 benchmarks, including EMNLP publications (e.g. [1,2,3]). This experimental design choice is common as social media-based stance detection data is difficult to collect at scale, leading to limited available datasets.
>
>
> > **Question (Q1.2):** [Need more] “in-depth analysis … I suspect the improvement of the performance may be due to the existence of certain stance-related cue words generated by chatgpt, rather than injecting world knowledge. The paper could be more informative if the authors could provide some analysis on the generated COT reasonings.”
>
> **Answer (A1.2):** We agree that the paper would be improved with a more in-depth analysis of why the COT reasonings are improving performance. However, we note that in the limited space allotted for a short paper, it was challenging to include additional analysis. We believe this work is well-suited for a short paper as our methods require only a brief section to explain in complete detail.
>
> Regarding your concern about world knowledge, we have manually checked a few samples. While some samples have no overt need for additional world knowledge, we do find numerous samples where ChatGPT does inject world knowledge not found in the original text. For further investigation, we will release all the LLM outputs as a part of our paper. Here is an example of a ChatGPT COT injecting world knowledge for your reference.
>
> > **Tweet:** Take hold of your Authority in Christ! Take back what the enemy has stolen! #authority #Life #Calling #SemST
>
> > **Label** *Against* Atheism
>
> > **ChatGPT COT:** Since this text mentions Christ and the enemy, it is likely coming from a Christian perspective. Therefore, the stance of the tweet is [NO], as it is promoting belief in a higher power.
>
> We additionally highlight the non-trivial nature of identifying the importance of stance-related cue words. Macro-level analysis of word-level contributions to performance would require significant human annotation. Given the difficulty of running such an experiment, we instead opt for extrinsic evaluation of COT reasonings via their ability to increase performance on two stance detection tasks. We, however, plan to release the outputs from our models for further analysis — making additional analysis feasible for future works.
>
> > **Question (Q1.3):** “I'm also not convinced that COT embedding could help implicit stance detection, since the authors only provide related analysis on one dataset (Twitter Stance).”
>
> **Answer (A1.3):** We appreciate your perspective on this matter. However, we might have different interpretations. We want to share additional details that might clarify our point. Implicit stance detection is required for both datasets evaluated in our study — namely Tweet-Stance and Presidential-Stance. We provide analysis of errors related to implicit stance detection on both datasets in sections 4.1 and 4.2. However, we appreciate that the example from the Presidential-Stance dataset in Appendix B does not highlight the need for implicit stance detection in this dataset. Thus, please consider an additional sample, which is In Favor of Joe Biden. We will add this example to the paper for clarity.
>
> >“@USER @USER and @USER have to sow doubt and cheat and will say after the debate that biden cheated - as they already are prior to. his cult will believe whatever he says. it’s so disturbing”
>
> Implicit in this sample is that Biden supporters view Trump supporters as 1) being members of a cult, 2) cheaters, and 3) willing to believe whatever Trump says. Such opinions are commonly held on Twitter amongst Biden supporters but difficult to ascertain from this sample alone.
>
> > **Question (Q2):** “[Results may suffer from] data leakage. It is possible that chatgpt has already seen these stance datasets during its training. It may be interesting to compare the results on other LLMs with chatgpt.”
>
> **Answer (A2):** Yes, you are right! Our results will be more sound if we can validate that the embeddings of COT-reasonings from other LLMs can improve baseline performance. Thus, we added additional experiments using two popular LLMs, Llama 2 [4] and Falcon [5], as they are high-performing open-source LLMs. The results of the additional experiments can be found in Table 1 and Table 2.  Recall that we use the Twitter-RoBERTa (TR) backbone to encode the tweet-only (TR-Tweet), the COT only (TR-COT), and both the Tweet and COT texts together (TR-Tweet+COT). Note that gpt-3.5-turbo results are from the original paper. LLama-2 and Falcon results are additional results run as part of the rebuttal.
>
> Our experiments confirm that with the recently released [llama-2-7b-chat](https://huggingface.co/meta-llama/Llama-2-7b-chat) our results hold for most experiments. Specifically, embedding Llama 2 COT reasonings gives us a **3.1 pt boost** on the Biden subset of Presidential-Stance and a **3.5 pt boost** on the Tweet-Stance dataset, respectively. However, we find that results do not improve with Llama2 COT embeddings on the Trump subset of the Presidential-Stance dataset. This however was expected as gpt-3.5-turbo, a model much more powerful than Llama2-7b-chat (which is evident from the results in [4]) only provides a minor improvement over the baseline TR-Tweet (Trump subset).
>
> We find that COT-reasonings from falcon-7b-instruct are less impactful, however this is expected as Falcon is significantly less powerful than Llama 2 (as indicated in [4]). *Thus, our additional experiments confirm that LLM performance is critical to the effectiveness of the proposed CoT based approach.* Please see the tables below for the full details of these additional experiments.
>
> |                | LLMs used to generate COT reasoning |  Biden      | Trump       |
> |----------------|--------------------------------------|-------------|-------------|
> | TR-Tweet       | -                                    | 73.9 (0.87) | 79.1 (0.59) |
> |   TR-COT       |             gpt-3.5-turbo            | 71.3 (1.28) | **81.5 (0.33)** |
> |                |            Llama-2-7b-chat           | 63.9 (0.49) | 69.1 (1.08) |
> |                |          Falcon-7b-instruct          | 53.0 (1.60) | 61.9 (1.11) |
> |   TR-Tweet+COT |             gpt-3.5-turbo            | **77.1 (1.46)** | 80.7 (0.61) |
> |                |            Llama-2-7b-chat           | 77.0 (0.47) | 78.4 (0.73) |
> |                |          Falcon-7b-instruct          | 76.0 (0.99) | 76.8 (0.19) |
>
> **Table 1:** F1 scores on Presidential-Stance using COT embeddings from 3 different LLMs. Due to the lack of development set in Presidential Stance,  results are reported as the mean of three experimental trials. The standard deviation is reported in parentheses.
>
>
> |                | LLMs used to generate COT reasoning | F1-score |
> |----------------|--------------------------------------|----------|
> | TR-Tweet       |                                      |   69.0   |
> |   TR-COT       |             gpt-3.5-turbo            | **75.7** |
> |                |            Llama-2-7b-chat           |   69.3   |
> |                |          Falcon-7b-instruct          |   66.5   |
> |   TR-Tweet+COT |             gpt-3.5-turbo            | **76.3** |
> |                |            Llama-2-7b-chat           |   72.5   |
> |                |          Falcon-7b-instruct          |   65.3   |
>
> **Table 2.**  F1 scores on Tweet-Stance using COT embeddings on 3 different LLMs. Unlike Presidential-Stance, Tweet-Stance has a development set, so we report results after early stopping as mentioned in Appendix A.
>
> We will share these new results in the main paper (as space permits) and/or in the appendix.
>
> >**Question (Q3):** “For the 1-shot COT proposed by the authors, how sensitive the approach is to the COT reasoning written by the authors?”
>
> **Answer (A3):** We did not do any prompt optimization for the 1-shot COT in this study. We simply picked a random training example and wrote an accurate COT to help the model understand the task. We hypothesize that analysis of prompt optimization can potentially boost performance even further.
>
> ---
>
> We appreciate your time and efforts to review our paper! We hope this response satisfies your concerns and resolves any potential ambiguity. Please let us know if our replies address your concerns.
>
> ----
>
> ## References
> [1] Yang Li and Jiawei Yuan. “Generative Data Augmentation with Contrastive Learning for Zero-Shot Stance Detection”. (EMNLP 2022)
>
> [2] Yingjie Li and Cornelia Caragea., Target-Aware Data Augmentation for Stance Detection. (NAACL 2021)
>
> [3] Matero, M., Soni, N., Balasubramanian, N., & Schwartz, H. A., MeLT: Message-level transformer with masked document representations as pre-training for stance detection. (EMNLP Findings 2021)
>
> [4] Touvron, H., Martin, L., Stone, K., Albert, P., Almahairi, A., Babaei, Y., ... & Scialom, T. (2023). Llama 2: Open foundation and fine-tuned chat models. arXiv preprint arXiv:2307.09288.
>
> [5] https://huggingface.co/tiiuae/falcon-7b-instruct

---

### Meta-Review · Area_Chair_bPCA · 2023-09-19

**Recommendation:** 3

**Metareview:**

The paper presents an approach for stance detection in social media, using Chain-of-Thought (COT) embeddings. The paper does not extract the label directly from the LM output. Instead, a BERT classifier is trained to use the LM output as an additional (or alternative) feature. Reviewers commented that the approach is simple and likely novel, and the idea could be relevant to a range of text classification tasks.

The main concerns were limited reproducibility because of the use of ChatGPT. Experimental comparison was also limited. Also, certain details such as data distribution were missing. Authors have provided additional experiments in the rebuttal for some of these concerns, and these should be added to the main paper.

---

### Decision · Program_Chairs · 2023-10-07

**Decision:**

Accept-Findings

**Comment:**

The paper presents an approach for stance detection in social media, using Chain-of-Thought (COT) embeddings. The paper does not extract the label directly from the LM output. Instead, a BERT classifier is trained to use the LM output as an additional (or alternative) feature. Reviewers commented that the approach is simple and likely novel, and the idea could be relevant to a range of text classification tasks.

The main concerns were limited reproducibility because of the use of ChatGPT. Experimental comparison was also limited. Also, certain details such as data distribution were missing. Authors have provided additional experiments in the rebuttal for some of these concerns, and these should be added to the main paper.